# How to Open a Black Box Classifier for Tabular Data

Bradley Walters, Sandra Ortega-Martorell, Ivan Olier * and Paulo J. G. Lisboa

School of Computer Science and Mathematics, Liverpool John Moores University, Liverpool L3 2AF, UK
* Correspondence: i.a.oliercaparroso@ljmu.ac.uk

**Abstract:** A lack of transparency in machine learning models can limit their application. We show that analysis of variance (ANOVA) methods extract interpretable predictive models from them. This is possible because ANOVA decompositions represent multivariate functions as sums of functions of fewer variables. Retaining the terms in the ANOVA summation involving functions of only one or two variables provides an efficient method to open black box classifiers. The proposed method builds generalised additive models (GAMs) by application of L1 regularised logistic regression to the component terms retained from the ANOVA decomposition of the logit function. The resulting GAMs are derived using two alternative measures, Dirac and Lebesgue. Both measures produce functions that are smooth and consistent. The term partial responses in structured models (PRiSM) describes the family of models that are derived from black box classifiers by application of ANOVA decompositions. We demonstrate their interpretability and performance for the multilayer perceptron, support vector machines and gradient-boosting machines applied to synthetic data and several real-world data sets, namely Pima Diabetes, German Credit Card, and Statlog Shuttle from the UCI repository. The GAMs are shown to be compliant with the basic principles of a formal framework for interpretability.

**Keywords:** ANOVA; Shapley values; self-explaining neural networks; generalised additive models; interpretability

## 1. Introduction

Machine learning models can be inherently interpretable, typically by fitting decision trees [1] or even by representing an existing black box model, such as a neural network, by extracting rules, whether using decompositional methods to explain the activity of individual hidden neurons, or applying pedagogical methods to fit the decision surface with axis-orthogonal hypercubes [2]. Decision trees have been successfully used to build transparent models in high-stakes applications [3].

Alternatively, statistical models, such as logistic regression, have high classification performance for the levels of noise typical for clinical prediction models [4]. Both decision trees and logistic regression have been successful and have global interpretability. However, each has a significant limitation. Rule sets can grow so complex as to become opaque to the user. Generalised linear models, while accurately modelling the nature of chance variation in the data through an appropriate choice of the output distribution function, require a priori choices of attribute factors, often resorting to categorising input variables to better capture non-linearities in the data.

A model that is arguably gold-standard should combine linear additivity with the automatic estimation of the non-linear dependence of the prediction on individual variables or pairs of variables. This can be achieved with generalised additive models (GAMs) [5]. We investigate to what extent it is possible to buck the performance-interpretability trade-off for tabular data by deriving GAMs from existing black box models, or using standard machine learning approaches to seed a GAM, keeping only univariate and bivariate terms.

Opening black boxes with ANOVA in this way is attractive because GAMs quantify Bayesian models in a way that is natural for human thinking. In particular, the representation of the logit as a GAM represents the prediction of the probability of class membership

as a combination of independent effects, much in the way that logistic regression does, but allowing for non-linear functions of the input variables. Specifically, for input dimensionality $d$, the odds ratio of this probability has the form

$$\frac{P(C|x)}{1 - P(C|x)} = e^{\varphi_1(x_1)}.e^{\varphi_1(x_2)} \ldots e^{\varphi_d(x_d)}.e^{\varphi_{1,2}(x_1, x_2)} \ldots e^{\varphi_{(d-1),d}(x_{(d-1)}, x_d)}.e^{\varphi_0} \tag{1}$$

where the terms $\varphi_i(x_i)$ are univariate functions, hence easily interpreted, $\varphi_{ij}(x_i, x_j)$ are bivariate functions, which can also be easily plotted, and $\varphi_0$ is the *null model* for which all of the input variables $\varphi_i(x_i)$ and $\varphi_{i,j}(x_i, x_j)$ are set to 0.

From a Bayesian perspective, each component $e^{\varphi}$ models the contribution of an individual variable or pair of variables, which can enhance or suppress the P(C | x) depending on whether the value taken by the argument function $\varphi$ is positive or negative, acting on the baseline value $e^{\varphi_0}$, which, if $\varphi_i(0)$ is always 0, corresponds to the prior odds ratio in the absence of any of the input variables being present.

As the variables are entered into Equation (1), they modulate the prediction of P(C | x), much in the same way as a human observer can start with a prior probability, e.g., for the diagnosis of a clinical state, then modulate that diagnosis as the observations of the symptoms are made; each symptom contributing to increasing or reducing the probability of diagnosing the clinical state according to Equation (1). In the medical domain, many risk models are quantified by exactly this model, usually expressed as a risk score, namely

$$score(x_1, x_2, \ldots, x_d) = \sum_{i=1}^{d} \beta_i \cdot x_i \tag{2}$$

with

$$logit(P(c|x)) = log\left(\frac{P(C|x)}{1 - P(C|x)}\right) = \beta_0 + score(x_1, x_2, \ldots, x_d) \tag{3}$$

This corresponds to the GAM defined by Equation (1) with only univariate terms given by

$$\varphi_i(x_i) = \beta_i \cdot x_i \tag{4}$$

### 1.1. Related Work on Self-Explaining Neural Networks

Interpretable neural network models have a long history starting with generalised additive neural networks (GANNs) [6–8], also called self-Explaining neural networks (SENNs) [9], which consist of a multilayer perceptron with modular structures that are not fully connected but involve sums of sub-networks, each representing functions of a single input variable or pair of variables [6]. However, they lack efficient methods to carry out a model selection to avoid modelling spurious correlations by including too many variables.

Our method relies on the analysis of variance (ANOVA) decompositions [10]. Although ANOVA is well known in mainstream statistics, its potential to derive interpretable models from pre-trained black box machine learning algorithms has not been fully exploited. In his paper introducing gradient boosting machines, Friedman notes that partial dependence functions can help "interpret models produced by any black box prediction method, such as neural networks, support vector machines, etc." [11]. However, this referred to the visualisation of the model's dependence on covariates, which applies locally only at the data median, rather than building a predictor that applies globally and so can be used to make predictions over the complete range of input values with the same additive model.

Other algorithms to derive predictive additive models have been proposed recently. They are neural additive models (NAM), where the univariate functions are each modelled with a separate multilayer perceptron or deep learning neural network [12] and explainable boosting machines (EBM) [13], which includes both univariate and bivariate terms. A recent refinement of these methods is the GAMI-NET [14]. This model estimates main (univariate) effects and pairwise interactions in two separate stages, building bespoke

neural networks to model each effect and interaction. None of these models will open an existing black box since they built a SENN structure first rather than applying function decomposition to a given multivariate function, as achieved with ANOVA.

Moreover, all the above models have limitations either in feature selection or in the structure of the model itself. In particular, NAMs favour a model structure that includes univariate functions for all the input variables and lack a clear process for selecting bivariate component functions. In contrast, EBMs incorporate model selection with statistical tests; in fact, ANOVA significance tests applied to partial dependence functions, proposed by [11], which are similar to the marginal functions used in Section 2.1.1 to calculate partial responses from ANOVA decompositions with the Lebesgue measure. This permits the inclusion of bivariate functions in the additive model. However, the EBM component functions are jagged because they are built from hyper box cuts in input space. The GAMI-NET requires explicit sparsity and heredity constraints, along with what is called marginal clarity, which is a penalisation term to enforce orthogonality between the main effects and the pairwise interactions. This is motivated by the functional ANOVA decomposition, implicitly using the marginal distribution, although it is not clear whether this observes the constraint raised in [11] to ensure that correlations among the input variables do not bias the orthogonality calculation. Our approach uses the ANOVA decomposition directly and so keeps the training process much simpler.

All of the above methods are stand-alone algorithms rather than explaining predictions made by pre-trained black boxes. This is also the case for sparse additive models (SAM) [15], where the component univariate and bivariate functions in a GAM are implemented with splines in contrast to our use of neural networks, which are semi-parametric, and hence, less restrictive. Moreover, splines can over-regularise the model and miss important details in the data, as well as being inefficient for estimating bivariate terms due to a proliferation of spline parameters. A further model, sparse additive machines [16], derives GAM structures from SVM models. It is scalable and has a provable convergence rate that is quadratic on the number of iterations, but this is not probabilistic and does not include pairwise terms.

The motivation for considering ANOVA as a method to open black box models is that each measure used in ANOVA is closely related to an intuitive approach for the decomposition of multivariate functions into predictive models with fewer variables. The Dirac measure filters from the multivariate response precisely the terms in the Taylor series, centred at the data median, which are dependent on just one or two variables [17], while the Lebesgue measure marginalises the response surface over one of two variables [18]. The proposed method is computationally efficient and stable for variable selection.

*1.2. Contributions to the Literature*

The main hypothesis of this paper is that the low-order functions derived by ANOVA from arbitrarily complex machine learning or other probabilistic classifiers contain sufficient information to open the black box models while retaining the classification performance measured by the area under the receiver operating characteristic curve (AUC). The resulting models are interpretable by design [19].

The proposed generic framework to extract GAMs from black boxes is termed partial responses in structured models (PRiSM). An instantiation of the framework has been demonstrated by applying the simplest measure used by ANOVA. This performed well on two medical data sets about intensive care [20] and heart transplants [21]. The focus of the latter is a detailed clinical interpretation of the partial response network (PRN), which is an anchored model, i.e., it combines functions restricted to be zero at the median values of the data. This paper makes a comprehensive approach to the proposed method and contributes novelty in the following respects:

- Comprehensive presentation of the generic framework for deriving PRiSM models from arbitrary black box binary classifiers, reviewing the orthogonality properties of ANOVA for two alternative measures: the Dirac measure, which is similar to partial dependence functions in visualisation algorithms [11] and produces component

functions that are tied to the data median; the Lebesgue measure, which involves estimates of marginal effects and is related to the quantification of effect sizes [7]. The method is tested on nine-dimensional synthetic data to verify that it retrieves the correct generating variables and achieves close to optimal classification performance;

- Derivation of a commonly used indicator of feature attribution, Shapley values [22]. When applied to the logit of model predictions from GAMs and SENNs, it is shown to be identical to the value of the contributions of the partial responses derived from ANOVA;

- Mapping of the properties of the PRiSM models to a formal framework for interpretability, demonstrating compliance with its main requirements [23], known as the three Cs of interpretability. This is complemented by an in-depth analysis of the component functions estimated from three real-world data sets.

The univariate and bivariate component functions representing the additive contributions to the logit of the model prediction are what we call partial responses. Note that while univariate component functions are numerically identical to partial dependence plots, the bivariate functions are not since they are obtained by removing the univariate dependence via the orthogonality properties of ANOVA decompositions. Moreover, the univariate and bivariate component functions are not used here purely for visualisation. Their values are the nomogram of the model, i.e., the ordinate of the figures shown later is in all cases the precise contribution of the variables to the model prediction, which is for every data point merely the summation of the contributions from all of the variables in the logit space.

The derived models retain a direct link between the input variables and the model predictions, meeting the requirements of the three "Cs" outlined earlier, and have comparable classification performance to the original black box models. This is demonstrated by application to four real-world data sets: UCI Diabetes, UCI German Credit Card, and Statlog Shuttle. The first three were used as benchmark data sets, whilst the last one was chosen as it was used in related work [12].

We refer to the overall framework to open pre-trained black boxes by deriving sparse models in the form of GAMs and SENNs as the integration of partial responses into structured models (PRiSM), Figure 1.

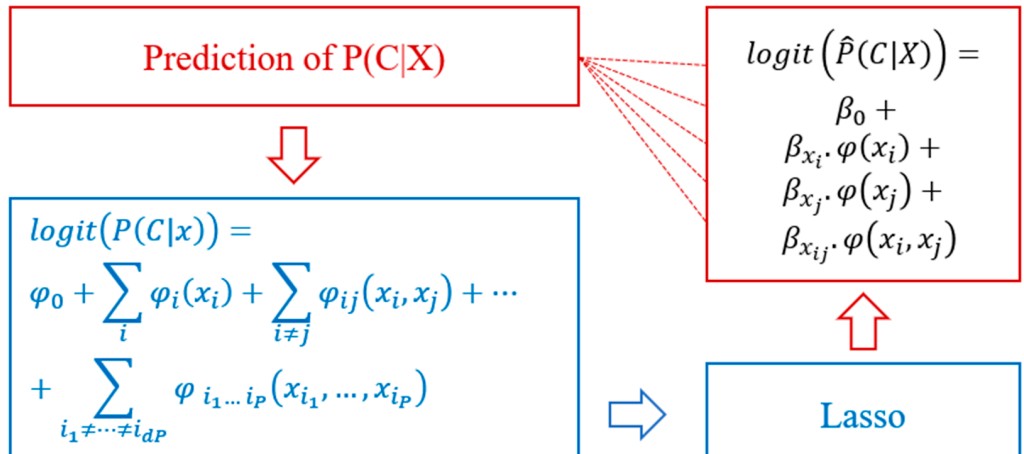

**Figure 1.** Schematic of the PRiSM framework. Any multidimensional decision function can be represented by a spectrum of additive functions, each with only one or two inputs. The final prediction of the probability of class membership, $\hat{P}(C|X)$, is given by the sum of the univariate and bivariate component functions, scaled by the coefficients $\beta_{x_i}$, $\beta_{x_{ij}}$ derived by the least absolute shrinkage and selection operator (LASSO). Since only univariate $\varphi(x_i)$ and bivariate $\varphi(x_i, x_j)$ component functions are in the model, their shapes provide a route towards interpretation by end-users.

## 2. Materials and Methods

### 2.1. Methods

#### 2.1.1. ANOVA Decomposition

The first novelty of the paper is to apply an ANOVA decomposition [10] to pre-trained black box probabilistic binary classifiers in order to extract from the *logit(P(C|x))*, which is a multivariate function, component functions of fewer variables.

The ANOVA decomposition is defined as follows

$$
\begin{aligned}
logit(P(C|x)) &\equiv \log\left(\frac{P(C|x)}{1-P(C|x)}\right) \\
&= \varphi_0 + \sum_i \varphi_i(x_i) + \sum_{i \neq j} \varphi_{ij}(x_i, x_j) + \ldots + \sum_{i_1 \neq \ldots \neq i_P} \varphi_{i_1 \ldots i_P}(x_{i_1}, \ldots, x_{i_P})
\end{aligned} \tag{5}
$$

where the general form of the terms in (5) is a recursive function of the nested subsets of the covariate indices $\{i_1, \ldots, i_P\}$ with the property that the term involving all of the covariates $x_i: i = 1 \ldots P$, where $P$ is the dimensionality of the input data is given by

$$
\begin{aligned}
&\varphi_{i_1 \ldots i_P}(x_{i_1}, \ldots, x_{i_P}) \\
&= logit\big(P(C|x_{i_1}, \ldots, x_{i_P})\big) - \sum_{\{i_1 \neq \ldots \neq i_{P-1}\}} \varphi_{i_1 \ldots i_{n-1}}(x_{i_1}, \ldots, x_{i_{P-1}}) - \varphi_0
\end{aligned} \tag{6}
$$

Note that Decomposition (5) is an identity that exactly reproduces the values of the $logit(P(C|x))$, originally predicted by the black box classifier. We call the component functions $\varphi_{i_1 \ldots i_n}(x_{i_1}, \ldots, x_{i_n})$ partial responses, since they involve only a subset of the input variables.

The general form of the component terms is given by the following equations, which depend only on the chosen measure $\mu(x)$

$$
\varphi_0 = \int_{[x]^P} logit(P(C|x))d\mu(x) \tag{7}
$$

$$
\varphi_S(x_s) = \int_{[x]^{P-|S|}} logit(P(C|x))d\mu(x_{-S}) - \sum_{T \subset S} \varphi_T(x_T) \tag{8}
$$

where $S \in R^S$ represents a subset of variables with dimensionality $|S| \leq P$. The terms $x_s$ and $x_{-s}$ denote, respectively, the subspace spanned by $S: |S| = n$ in Equation (6) and its complement $-S: |-S| = d - n$.

It follows from (7) and (8) that the terms $\varphi_S$ are normalised with respect to the chosen measure

$$
\int_S \varphi_S(x_s)d\mu(x_j) = 0, \; if \; j \in S \tag{9}
$$

and also orthogonal for disjoint variable sets $S$ and $T$

$$
\int_S \varphi_S(x_s)\varphi_T(x_T)d\mu(x) = 0, \; if \; S \neq T. \tag{10}
$$

There are two natural choices of measure, each of which will define the functionality of each of the component terms $\varphi_S$ in response to either one or two arguments:

- Dirac measure

$$
d\mu(x) = \delta(x - x_c)dx \tag{11}
$$

An arbitrary point $x_c$ that is called anchor point. The partial responses become cuts through the response surface for the *logit(P(C|x)*.

- Lesbesgue measure

$$
d\mu(x) = \rho(x)dx \tag{12}
$$

where $\rho(x)$ is the density function of the variables in the argument of the integral. This measure calculates the weighted mean of the integrand.

In both cases, the data matrix $X$ is first centred using the overall median of the data and scaled by the marginal standard deviation:

$$X \rightarrow (X - median(X)) \Big/ std(X) \tag{13}$$

The absence of a variable now corresponds to fixing it at the median value, since the median point corresponds to a vector of 0 s. Therefore, the logit value then takes the value $logit(P(C|0))$. Similarly, if all of the variables except $x_i$ are set to their median values, then the corresponding values of $logit(P(C|(0,\dots,x_i,\dots,0)))$ represent a function of just that one variable. The same principle applies when only two variables are not 0, then three variables, etc.

The partial responses for the Dirac measure are calculated according to

$$\varphi_0 = logit(P(C|0)) \tag{14}$$

$$\varphi_i(x_i) = logit(P(C|(0,\dots,x_i,\dots,0))) - \varphi_0 \tag{15}$$

$$\varphi_{ij}(x_i, x_j) = logit(P(C|(0,\dots,x_i,\dots,x_j,\dots,0))) - \varphi_i(x_i) - \varphi_j(x_j) - \varphi_0 \tag{16}$$

In the case of the Lebesgue measure, the integrals in Equations (7) and (8) are calculated empirically using the training data, with sample size $N$ observations

$$\hat{F}_S(x_s) = \frac{1}{N} \sum_{k=1}^{N} logit\left(P(C|x_S, x_{-S}^k)\right) \tag{17}$$

where the variables with dimensions $x_s$ take any desired values but those in the complement set with dimension $x_{-S}^k$ are fixed at their actual values in the training set $k = 1 \dots N$ [11]. This corresponds to shifting all onto the coordinate(s) $x_s$ so that in the summation (17), every data point has the same value of this input dimension while retaining the original values for all other coordinates.

The orthogonalised partial responses $\varphi_S(x_s)$ follow by using Equation (8).

$$\hat{\varphi}_0 = \frac{1}{N} \sum_{k=1}^{N} logit\left(P(C|x^k)\right) \tag{18}$$

$$\hat{\varphi}_i(x_i) = \hat{F}_i(x_i) - \hat{\varphi}_0 \tag{19}$$

$$\hat{\varphi}_{ij}(x_i, x_j) = \hat{F}_{ij}(x_i, x_j) - \hat{\varphi}_i(x_i) - \hat{\varphi}_j(x_j) - \hat{\varphi}_0 \tag{20}$$

2.1.2. Model Selection with the LASSO

The resulting terms in the truncated ANOVA decomposition comprising only the univariate and bivariate terms in (5) need to be re-calibrated to maximise the predictive classification performance. In addition, having treated the higher-order terms as noise, the remaining terms need also to be filtered to remove non-informative partial responses.

This is achieved through a second step involving the application of the logistic regression with the LASSO [24], treating $P * (P+1)/2$ terms in the truncated ANOVA decomposition as the new input variables. The $L_1$ regularisation is robust for hard model selection by sliding to zero the value of the linear coefficients for the least informative variables, which are now partial responses.

Since the partial responses are generally non-linear functions of one of two variables, they are readily interpretable. This is not new having previously been a widely used approach to visualise non-linear models with partial dependence functions that are func-

tionally equivalent to ANOVA terms with the Dirac measure. What is new is the realisation that when these functions are calculated for the *logit(P(C | x)* and used for prediction with tabular data, they can achieve comparable AUCs to those of the original black box models from which the partial responses are derived.

2.1.3. Second Training Iteration

If the original black box model is an MLP, it is possible to construct a GANN/SENN to replicate the output of the logistic Lasso by a replication of the weights from the MLP multiplied by the coefficients of the Lasso. This permits an additional step of refining the partial responses themselves by initialising the SENN at the operating point of the GAM.

Given the weights $\{w_{ij}, b_j, v_j, v_0\}$ of the original pre-trained fully connected MLP, Figure 2, with the inputs indexed by $i$ and hidden nodes indexed by $j$, together with the co-efficients $\{\beta_0, \beta_i, \beta_{ij}\}$ fitted by the Lasso, the PRiSM model for the anchored decomposition can be exactly mapped onto an MLP structured in the form of a GANN/SENN, as follows.

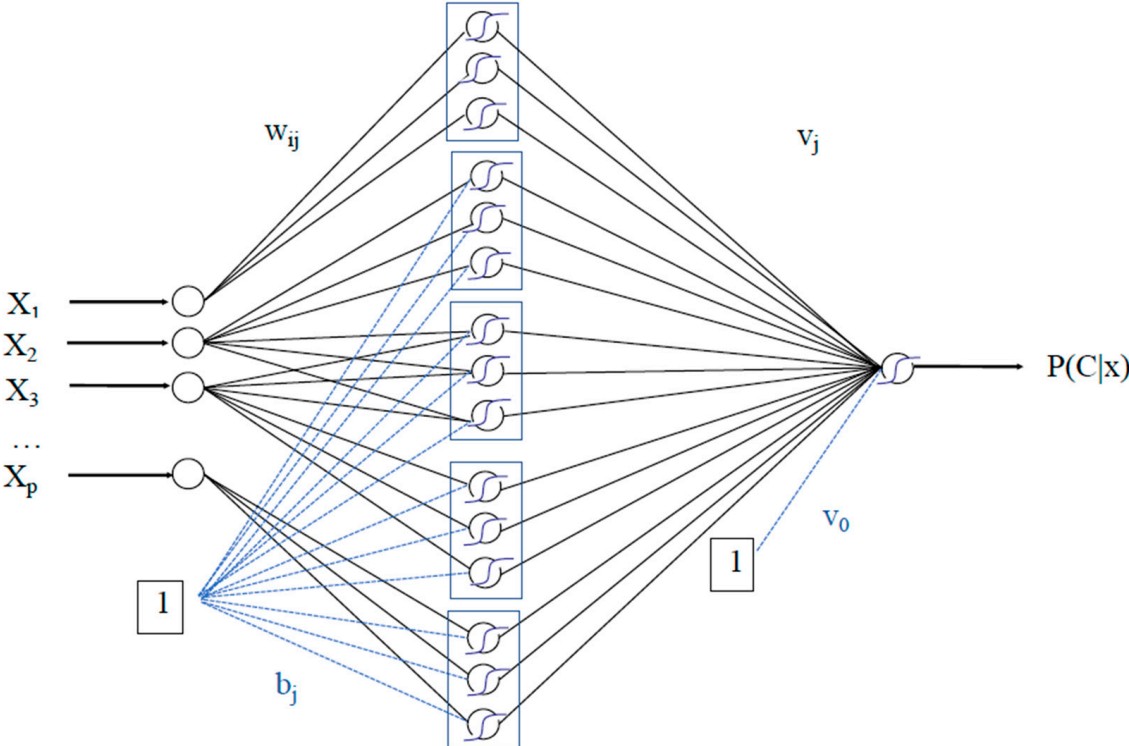

**Figure 2.** The partial response network (PRN) has the modular structure typical of a self-explaining neural network. In particular, the figure illustrates the connectivity for a univariate function of input variables $x_1$ and $x_p$, and a bivariate term involving variables $x_2$ and $x_3$. Modelling the interaction term as orthogonal to univariate terms involving the same variables requires three blocks of hidden nodes, as explained in the main text. If there are univariate additive component functions involving either $x_2$ or $x_3$ these are added to the structure by inserting additional modules, as shown for $x_1$ and $x_p$.

(1)　　Univariate partial response corresponding to the input $x_i$

This is shown in Figure 2 for input $x_1$. Zero inputs for all other inputs will not contribute to the activation of the hidden nodes. The hidden layer weights $w_{ij}$ connected to node $x_i$ remain the same as in the original MLP but the weights and bias to the output node need to be adjusted as follows:

$$v_j \to \beta_i * v_j \qquad (21)$$

$$v_0 \rightarrow \beta_i * (v_0 - logit(P(C|0))) \tag{22}$$

(2)　Bivariate partial response for the input pair $\{x_i, x_j\}$

This is shown in Figure 2 for inputs $x_2$ and $x_3$. This time, to replicate the partial response multiplied by the Lasso coefficient, it is necessary to add three elements to the structure, namely, a univariate partial response for each of the inputs involved and a coupled network that both inputs feed into together. We will use the generic input indices $k$ and $l$ to avoid confusion with the hidden node index $j$. The hidden layer weights once again remain unchanged from the original MLP. The output layer weights and bias for the network structure representing the univariate term associated with input $k$ (and similarly for input $l$) are adjusted by

$$v_j \rightarrow (\beta_k - \beta_{kl}) * v_j \tag{23}$$

$$v_0 \rightarrow (\beta_k - \beta_{kl}) * (v_0 - logit(P(C|0))) \tag{24}$$

whereas the weights and bias for the coupled network are changed according to

$$v_j \rightarrow \beta_{kl} * v_j \tag{25}$$

$$v_0 \rightarrow \beta_{kl} * (v_0 - logit(P(C|0))). \tag{26}$$

(3)　Finally, an amount is added to the total sum of the values calculated for the bias term in the structured neural network. This amount is equal to the intercept of the logistic Lasso, $\beta_0$.

Equations (23)–(26) have the property that when an interaction between two variables is identified after the first application of the Lasso, its mapping onto the structured neural network involves also univariate modules, with the consequence that the second back-propagation step can potentially render the bivariate term not statistically significant and replace it with one or more univariate independent effects.

2.1.4. Summary of the Method

The following is the pseudo-code for PRiSM models, taking as a starting point a data set comprising a set of P-dimensional input variables normalised according to (13) and the corresponding black box model predictions.

Algorithm Partial Response Models *PRiSM(BB, D):*

**Input:** set $D$ of training examples; predictions $P(C|x)$ from a pre-trained black box model *BB*.

**1. ANOVA decomposition:** apply the recursion given by Equation (6) to the *logit(P(C|x)*.

This may use either of the suggested measures, Dirac and Lesbesgue. The former leads to an anchored decomposition referenced to the choice of anchor point; the component functions generated by the latter are summations weighted by the density function of the covariates, *P(x)*.

**2. Model selection with the Lasso**: input the set of univariate and bivariate partial responses $\varphi_i(x_i)$ and $\varphi_{ij}(x_i, x_j)$ from (6) calculated over the training data set $D$ as new inputs for variable selection with the logistic regression Lasso.

The Lasso will also output a linear coefficient for each partial response, $\beta_i$ *and* $\beta_{ij}$, as well as an intercept $\beta_0$, generally resulting in good calibration.

**Output *prBB(BB, D)*:** this is the output of the Lasso in Step 2, which has the form of a GAM, shown in Equation (5), truncated to the selected subset of functions $\varphi_i(x_i)$ and $\varphi_{ij}(x_i, x_j)$:

$$logit(prBB(C|x)) \equiv \varphi(0) + \sum_i \beta_i \varphi_i(x_i) + \sum_{i \neq j} \beta_{ij} \varphi_{ij}(x_i, x_j) \tag{27}$$

Each partial response comprises a non-linear function of its arguments. Consequently, the model prediction equals the sum of all partial responses plus the intercept, weighted by the linear coefficients from Step 2, followed by the application of the sigmoid function, which inverts the *logit(P(C | x))*.

The predictions for anchored decompositions are indexed by the pre-fix *pr* followed by an abbreviation of the black box algorithm, e.g., prSVM and prGBM.

**3. Predictions with *PRiSM models*:** given a test data point, the $\varphi_i(x_i)$ and $\varphi_{ij}(x_i, x_j)$ are calculated using Equations (14)–(16) or (17)–(20), and the predicted output follows from (27). The input variables are, therefore, directly linked to the predictions through interpretable functions.

Anchored decomposition applied to the MLP:

By denoting the output *prMLP (MLP, D)* by *MLP-Lasso*, it is then possible to continue training as follows:

**4. Map the MLP-Lasso into a GANN/SENN**: this has the form of a GANN/SENN, meaning that it is not fully connected, as shown in Figure 2. The adjustments to the weights are explained in Equations (21)–(26) and in Section 2.1.3.

**Output Partial Response Network [PRN]:** having initialised a structured neural network in the previous step, so that it exactly replicates the component functions and output of the MLP-Lasso, back-error propagation is applied to continue the training of this network. The PRN is a probabilistic binary classifier, so training will use the log-likelihood cost function. Note that the component functions no longer conform with the requirements of an ANOVA decomposition as they will have been adjusted without the constraint of orthogonality.

**Output PRN-Lasso:** Steps 1,2. are then applied to the PRN instead of the original MLP. This generates a new set of partial responses $\varphi_i^*(x_i)$ and $\varphi_{ij}^*(x_i, x_j)$ and corresponding coefficients $\beta_i^*$ *and* $\beta_{ij}^*$ from which the model predictions follow by inserting these coefficients and partial responses into Equation (27).

The second training iteration enables the partial responses to be refined without being adjusted for input variables that were removed from the model by the Lasso in Step 2. Therefore, the PRN–Lasso–BB models will always comprise a subset of the variables in the PRN–BB models, but not necessarily with the same functional form. It is possible that some of the partial responses in the PRN–BB model are no longer selected and even that what may have started as a pure two-way effect can now be split into two independent main effects represented by univariate partial responses.

2.1.5. Exact Calculation of Shapley Values

In common with GAMs generally, the interpretation of a PRiSM model is the model itself, since the components of the logit are additive and the amount contributed by each partial response is clearly quantified.

In addition, the relevance of individual input variables can be calculated using Shapley values [22]. This can be achieved for the overall prediction of the probability of class membership, but also for the logit, which places less emphasis on the non-linearity at the class boundary but takes greater account of the value of the logit across the full range of input values, which is important to ensure the good calibration of the final model.

When the logit has the form shown in Equation (27), the Shapley value $\phi_i$ corresponding to input variable $x_i$ of dimensionality $P$ can be efficiently computed by summing over all variable subsets that exclude input $i$, $S \subseteq P\backslash\{i\}$, including $S = \varnothing$, with the usual formula

$$\phi_i(x_i) = \frac{1}{P} \sum_{S \subseteq N\backslash\{i\}} \binom{P-1}{|S|}^{-1} (logit(P(C|S \cup \{i\}) - logit(P(C|S)) \tag{28}$$

The linear terms in (27) simply add for every combination of input variables excluding *i*, of which there are $\binom{P-1}{|S|}$; therefore, at each data point the contribution to $\phi_i(x_i)$ is just

$\beta_i(\varphi_i(x_i) - \varphi_i(0))$. In the case of the bivariate terms, the calculation is similar but involves only $\begin{pmatrix} P-2 \\ |S| \end{pmatrix}$ combinations giving

$$\frac{1}{P}\sum_{j=1}^{P-1}\left[\frac{\begin{pmatrix} P-2 \\ j-1 \end{pmatrix}}{\begin{pmatrix} P-1 \\ j \end{pmatrix}}\right] = \frac{1}{P}\sum_{j=1}^{P-1}\left[\frac{(P-2)!j!}{(P-1)!(j-1)!}\right] = \frac{1}{P}\sum_{j=1}^{P-1}\left(\frac{j}{P-1}\right) = \frac{1}{2} \tag{29}$$

Therefore, the pairwise terms neatly share their impact among the Shapley values for each of the variables, yielding

$$\phi_i(x_i) = \beta_i(\varphi_i(x_i) - \varphi_i(0)) + \frac{1}{2}\sum_j \beta_{ij}(\varphi_{ij}(x_i, x_j) - \varphi_{ij}(0, x_j)) \tag{30}$$

2.1.6. Experimental Settings

All algorithms were implemented in Matlab [25]. The MLP was trained with automatic relevance determination [26] implemented in Netlab [27], although conjugate gradient descent leads to similar results. The other machine learning algorithms predict scores for a class assignment. In this study, the scores are calibrated for probabilistic classification using the score as the sole input to a logistic regression model, which forms the starting point for PRiSM models by taking the logit in the same way as for the MLP. The SVM was fitted using *fitcSVM* with an RBF kernel and automatic optimisation. The GBM model is implemented with *fitcensemble*, which boosts 100 decision trees using the function *LogitBoost*.

*2.2. Data sets used*

2.2.1. Synthetic Data

(a) <u>2D circle</u>. We implemented a sample size of n = 10,000 with unbalanced classes, which is the case for all synthetic data sets in this paper. In this case, the *logit* has two separate univariate components,

$$logit(P(C|(x_1, x_2)) = 10 \times \left[\left(x_1 - \frac{1}{2}\right)^2 + \left(x_2 - \frac{1}{2}\right)^2 - 0.08\right] \tag{31}$$

This data set is similar to that used in [15] but instead of generating clean data by allocating different classes on either side of the boundary, we use noisy data by generating binary targets with a Bernoulli distribution, which is also common for all the synthetic data sets that we report,

$$Y \sim Bin(n, P(C|(x_1, x_2))). \tag{32}$$

The factor of 10 in Equation (31) is to reduce the amount of noise and so ensure a reasonable value for the AUC. The values of $(x_1, x_2)$ are generated using $x_i = 0.5 \times (u_i + w)$, where both $u_i$ and $w$ are uniform distributions in the range [0,1], to demonstrate the prediction accuracy when the two input variables are correlated. There are only two univariate main effects and no interaction term.

(b) <u>XOR function.</u> The purest bivariate interaction is the XOR, represented in the multilinear form appropriate for continuous Boolean algebra [28],

$$P(C|(x_3, x_4)) = x_3 + x_4 - 2x_3x_4, x_i \in ]0, 1[ \tag{33}$$

Each variable will be generated by a uniform distribution in [0,1]. This density function has the property that

$$logit(P(C|(x_3, x_4))) = log\left(\frac{x_3 + x_4 - 2x_3x_4}{1 - x_3 - x_4 + 2x_3x_4}\right) \tag{34}$$

Therefore, $logit\left(P\left(C\middle|\left(x_3, \frac{1}{2}\right)\right)\right) = 0$ making it a pure interaction for the ANOVA decomposition with the *Dirac measure* anchored at (1/2,1/2). Similarly, for the *Lebesgue measure*, it is readily shown that

$$logit(P(C|(x_3, x_4))) = -logit(P(C|(1 - x_3, x_4))) \tag{35}$$

which is similar to the other dimension; therefore, the integrals corresponding to the univariate terms vanish, once again leaving the pure interaction term.

(c) Logical AND function. A combination of univariate and bivariate terms by generating data according to the logical AND function, which in continuous Boolean algebra is represented by the following atomic term:

$$P(C|(x_5, x_6)) = x_5 x_6, x_i \in ]0, 1[ \tag{36}$$

This time, the ANOVA expansion with the *Dirac measure* anchored at (1/2,1/2) is

$$\begin{aligned} logit(P(C|(x_5, x_6))) \\ = -\log(3) + \sum_i \left[\log\left(\frac{x_i}{2 - x_i}\right) + \log(3)\right] \\ + \left[\log\left(\frac{(2 - x_5)(2 - x_6)}{1 - x_5 x_6} - \log(3)\right)\right] \end{aligned} \tag{37}$$

The *Lebesgue measure* yields univariate terms given by

$$\varphi_i^{Lebesgue} = \log\left(x_i / (1 - x_i)\right) + \frac{\log(1 - x_i)}{x_i} + Li_2(1) \tag{38}$$

where $Li_2(1)$ is the polylogarithm function of second order evaluated at 1. The bivariate term is given by the explicit ANOVA decomposition in Equation (5) and does not reduce to a simpler algebraic form.

(d) Three-way interaction. The final synthetic data set comprises a data set that cannot be modelled with univariate and bivariate terms only. The purpose is now to see how well PRiSM models work to model a high-order effect.

$$P(C|(x_7, x_8, x_9)) = x_7 x_8 x_9, x_i \in ]0, 1[ \tag{39}$$

Note that the complete set of input variables for the synthetic data set is calculated only once. In this way, the same sample of 9-dimensional input data will be used for all classifiers. Only the target classes differ, thus generating four separate binary classification tasks. In each case, two or three variables will carry signal and the others comprise noise. A minimum requirement of all classifiers is to identify the relevant input dimensions and discard the rest. In addition, since we have the data generators, we can calculate the optimal classification performance corresponding to allocating every data point to the correct class, irrespective of the stochastic label generated by the Bernoulli distribution.

2.2.2. Real-World data

A description of the variables included in the starting pool for model selection and any standardisation that was applied to them is provided below.

(a)　Diabetes data set:

The Diabetes dataset [29,30] comprises measurements recorded from 768 women, who were at least 21 years old, of Pima Indian heritage, and tested for diabetes using World Health Organization criteria. One of the variables, "Blood Serum" Insulin, has significant amounts of missing data. These rows were removed along with all entries with missing values of "Plasma Glucose Concentration" in a tolerance test, "Diastolic Blood Pressure" (BP), "Triceps Skin Fold Thickness" (TSF) or "Body Mass Index" (BMI), resulting in a reduced data set with n = 532. In line with common practice, a subset was randomly

selected for training (n = 314), and the remaining were used for testing (n = 268). The additional variables available are "Age", "Number of Pregnancies", and "Diabetes Pedigree Function" (DPF), a measure of family history of diabetes. A binary target variable indicated diabetes status, with a positive prevalence of 35.7%.

(b)     Statlog German Credit Card data set:

We used the numerical version of the Statlog German Credit Card database [31], which contains n = 1000 instances and 24 attributes. The first 700 observations were used for training, with a prevalence of bad credit risks being 29.6%. The remaining 300 observations were used for testing, with a prevalence of bad risks of 31%. The data set was used in the form created for the benchmarking study Statlog, where three categorical variables ("Other Debtors", "Housing", and "Employment") were coded in binary form with multiple columns.

(c)     Statlog Shuttle data set:

The Statlog Shuttle database [31,32] from NASA comprises 9 numerical attributes and an outcome label. It is split into 43,500 cases for training and 14,500 for testing. There are 7 outcomes, of which 21% are in the category "Rad Flow". The binary classification task is to separate this category, Class 0, from the others, assigned to Class 1. Given the strong imbalance between classes, the default accuracy for a null model, i.e., predicting the predominant class for all rows, is 79%. The target accuracy is 99–99.9%.

## 3. Results

The following sections compare the performance and characteristics of different PRiSM models obtained by opening a range of frequently used black box algorithms.

### 3.1. Synthetic Data

The purpose of the benchmarking on the synthetic data is to ascertain how close each machine learning classifier and the corresponding interpretable PRiSM models get to the optimal classification accuracy, which is obtained using the known class membership probabilities given by the generating formulae for class membership, notwithstanding the presence of noise in the targets.

The classification performance of frequently used machine learning models and their interpretable versions applied to the four synthetic sets are listed in Tables 1–4. The optimal AUC values are in bold, and the values below the confidence interval (CI) are in italics. Two-dimensional plots of the relevant variables from the nine input dimensions are plotted in Figures 3–5, showing the actual training data with Bernoulli noise and the ideal class allocations used to find the best achievable AUC. The generated data set was split into three parts for training, model parameter optimisation with out-of-sample data, and performance estimation in generalisation. It is interesting to see how much the optimal AUC varies between three slices from the same noisy data. This illustrates the importance of calculating confidence intervals. The model with marginally the best point estimate of the AUC for the optimisation data may not have the highest AUC estimated on the independent sample.

**Table 1.** Classification performance for the 2D circle measured by the AUC [CI]. The input variables $x_1$ and $x_2$ are ideally selected solely for their univariate responses.

| AUC [CI] | No. Input Variables | Training (n = 6000) | Optimisation (n = 2000) | Performance Estimation (n = 2000) |
|---|---|---|---|---|
| **Optimal classifier** | **2** | **0.676 [0.662,0.689]** | **0.657 [0.634,0.681]** | **0.666 [0.643,0.690]** |
| MLP | 9 | 0.676 [0.663,0.690] | 0.659 [0.635,0.682] | 0.660 [0.636,0.684] |
| SVM | 9 | 0.695 [0.682,0.708] | 0.646 [0.622,0.670] | 0.648 [0.624,0.672] |
| *GBM* | *9* | *0.697 [0.684,0.710]* | *0.649 [0.625,0.673]* | *0.641 [0.617,0.665]* |

**Table 1.** *Cont.*

| AUC [CI] | No. Input Variables | Training (n = 6000) | Optimisation (n = 2000) | Performance Estimation (n = 2000) |
|---|---|---|---|---|
| *PRiSM models* | *Components* | | *Dirac measure* | |
| Lasso | 2 | 0.675 [0.661,0.688] | 0.658 [0.634,0.682] | 0.661 [0.637,0.685] |
| PRN | 2 | 0.676 [0.662,0.689] | 0.659 [0.636,0.683] | 0.664 [0.640,0.687] |
| PRN–Lasso | 2 | 0.676 [0.662,0.689] | 0.659 [0.636,0.683] | 0.664 [0.640,0.688] |
| prSVM | 2 | 0.676 [0.662,0.689] | 0.658 [0.634,0.681] | 0.664 [0.640,0.688] |
| prGBM | 5 | 0.681 [0.667,0.694] | 0.655 [0.631,0.679] | 0.655 [0.632,0.679] |
| *PRiSM models* | *Components* | | *Lebesgue measure* | |
| Lasso | 2 | 0.675 [0.662,0.689] | 0.659 [0.636,0.683] | 0.661 [0.637,0.685] |
| PRN | 2 | 0.676 [0.662,0.689] | 0.659 [0.636,0.683] | 0.664 [0.640,0.687] |
| PRN–Lasso | 2 | 0.676 [0.662,0.689] | 0.660 [0.636,0.683] | 0.664 [0.640,0.687] |
| prSVM | 3 | 0.675 [0.662,0.689] | 0.657 [0.634,0.681] | 0.665 [0.641,0.689] |
| prGBM | 2 | 0.673 [0.659,0.686] | 0.656 [0.632,0.679] | 0.654 [0.630,0.678] |

**Table 2.** Classification performance for the XOR function measured by the AUC [CI]. The input variables $x_3$ and $x_4$ are ideally selected solely for their bivariate response.

| AUC [CI] | No. Input Variables | Training (n = 6000) | Optimisation (n = 2000) | Performance Estimation (n = 2000) |
|---|---|---|---|---|
| **Optimal classifier** | **1** | **0.689 [0.675,0.702]** | **0.663 [0.639,0.687]** | **0.671 [0.648,0.695]** |
| MLP | 9 | 0.692 [0.678,0.705] | 0.665 [0.641,0.688] | 0.669 [0.646,0.693] |
| SVM | 9 | 0.708 [0.695,0.721] | 0.652 [0.628,0.676] | 0.660 [0.637,0.684] |
| GBM | 9 | 0.713 [0.700,0.726] | 0.586 [0.561,0.610] | 0.609 [0.584,0.633] |
| *PRiSM models* | *Components* | | *Dirac measure* | |
| Lasso | 1 | 0.688 [0.675,0.701] | 0.663 [0.639,0.686] | 0.672 [0.648,0.695] |
| PRN | 1 | 0.690 [0.677,0.703] | 0.664 [0.640,0.687] | 0.670 [0.646,0.694] |
| PRN–Lasso | 1 | 0.688 [0.675,0.702] | 0.663 [0.639,0.686] | 0.672 [0.648,0.695] |
| prSVM | 14 | 0.691 [0.678,0.705] | 0.663 [0.640,0.687] | 0.671 [0.648,0.695] |
| prGBM | 1 | 0.687 [0.674,0.700] | 0.656 [0.633,0.680] | 0.661 [0.638,0.685] |
| *PRiSM models* | *Components* | | *Lebesgue measure* | |
| Lasso | 1 | 0.689 [0.676,0.702] | 0.664 [0.640,0.688] | 0.670 [0.647,0.694] |
| PRN | 1 | 0.690 [0.677,0.703] | 0.664 [0.640,0.687] | 0.670 [0.646,0.693] |
| PRN–Lasso | 1 | 0.690 [0.676,0.703] | 0.664 [0.641,0.688] | 0.670 [0.647,0.694] |
| prSVM | 7 | 0.690 [0.677,0.703] | 0.633 [0.640,0.687] | 0.672 [0.648,0.695] |
| prGBM | 1 | 0.688 [0.675,0.702] | 0.656 [0.632,0.680] | 0.659 [0.635,0.682] |

**Table 3.** Classification performance for the logical AND function measured by the AUC [CI]. The input variables $x_5$ and $x_6$ are ideally selected with two univariate responses and a bivariate response.

| AUC [CI] | No. Input Variables | Training (n = 6000) | Optimisation (n = 2000) | Performance Estimation (n = 2000) |
|---|---|---|---|---|
| **Optimal classifier** | **3** | **0.816 [0.802,0.830]** | **0.836 [0.813,0.860]** | **0.817 [0.793,0.841]** |
| MLP | 9 | 0.816 [0.803,0.830] | 0.833 [0.809,0.857] | 0.815 [0.791,0.839] |
| *SVM* | *9* | *0.803 [0.790,0.817]* | *0.797 [0.772,0.821]* | *0.786 [0.762, 0.809]* |
| GBM | 9 | 0.822 [0.810,0.834] | 0.826 [0.805,0.847] | 0.808 [0.787,0.830] |
| *PRiSM models* | *Components* | | *Dirac measure* | |
| Lasso | 3 | 0.815 [0.801,0.828] | 0.833 [0.809,0.857] | 0.813 [0.789,0.837] |
| PRN | 3 | 0.816 [0.802,0.829] | 0.835 [0.811,0.858] | 0.814 [0.790,0.838] |
| PRN–Lasso | 3 | 0.816 [0.802,0.830] | 0.835 [0.811,0.859] | 0.814 [0.791,0.838] |
| prSVM | 6 | 0.800 [0.787,0.813] | 0.813 [0.790,0.835] | 0.797 [0.774, 0.820] |
| prGBM | 6 | 0.820 [0.807,0.832] | 0.828 [0.807,0.848] | 0.807 [0.786,0.829] |

<table>
<tr><td colspan="5" align="center">Table 3. *Cont.*</td></tr>
</table>

| AUC [CI] | No. Input Variables | Training (n = 6000) | Optimisation (n = 2000) | Performance Estimation (n = 2000) |
|---|---|---|---|---|
| *PRiSM models* | *Components* | | *Lebesgue measure* | |
| Lasso | 3 | 0.815 [0.801,0.828] | 0.832 [0.808,0.856] | 0.813 [0.789,0.837] |
| PRN | 3 | 0.816 [0.802,0.829] | 0.835 [0.811,0.858] | 0.814 [0.790,0.838] |
| PRN–Lasso | 3 | 0.816 [0.802,0.830] | 0.835 [0.811,0.858] | 0.815 [0.791,0.839] |
| prSVM | 4 | 0.799 [0.786,0.812] | 0.812 [0.790,0.834] | 0.796 [0.773,0.819] |
| prGBM | 8 | 0.817 [0.805,0.829] | 0.828 [0.808,0.849] | 0.810 [0.789,0.831] |

**Table 4.** Classification performance for the three-way interaction measured by the AUC [CI]. Three input variables are involved, $x_7$, $x_8$, and $x_9$.

| AUC [CI] | No. Input Variables | Training (n = 6000) | Optimisation (n = 2000) | Performance Estimation (n = 2000) |
|---|---|---|---|---|
| **Optimal classifier** | **3** | **0.840 [0.822,0.859]** | **0.817 [0.783,0.851]** | **0.836 [0.805,0.868]** |
| MLP | 9 | 0.840 [0.822,0.859] | 0.809 [0.775,0.843] | 0.832 [0.801,0.864] |
| *SVM* | *9* | *0.797 [0.779,0.815]* | *0.764 [0.729,0.798]* | *0.786 [0.755,0.817]* |
| GBM | 9 | 0.831 [0.816,0.847] | 0.796 [0.767,0.826] | 0.813 [0.786,0.840] |
| *PRiSM models* | *Components* | | *Dirac measure* | |
| Lasso | 3 | 0.837 [0.818,0.855] | 0.811 [0.777,0.845] | 0.821 [0.797,0.861] |
| PRN | 3 | 0.837 [0.819,0.856] | 0.812 [0.778,0.846] | 0.830 [0.799,0.862] |
| PRN–Lasso | 3 | 0.837 [0.819,0.856] | 0.812 [0.778,0.846] | 0.830 [0.799,0.862] |
| prSVM | 6 | 0.813 [0.796,0.829] | 0.777 [0.744,0.810] | 0.807 [0.778,0.836] |
| prGBM | 3 | 0.832 [0.817,0.847] | 0.797 [0.768,0.826] | 0.813 [0.786,0.841] |
| *PRiSM models* | *Components* | | *Lebesgue measure* | |
| Lasso | 3 | 0.834 [0.816,0.853] | 0.808 [0.774,0.842] | 0.828 [0.796,0.860] |
| PRN | 3 | 0.837 [0.819,0.856] | 0.812 [0.778,0.846] | 0.831 [0.799,0.862] |
| PRN–Lasso | 3 | 0.837 [0.819,0.856] | 0.812 [0.778,0.846] | 0.831 [0.799,0.862] |
| prSVM | 6 | 0.808 [0.792,0.824] | 0.776 [0.745,0.808] | 0.805 [0.777,0.833] |
| prGBM | 4 | 0.825 [0.809,0.841] | 0.798 [0.768,0.828] | 0.810 [0.781,0.839] |

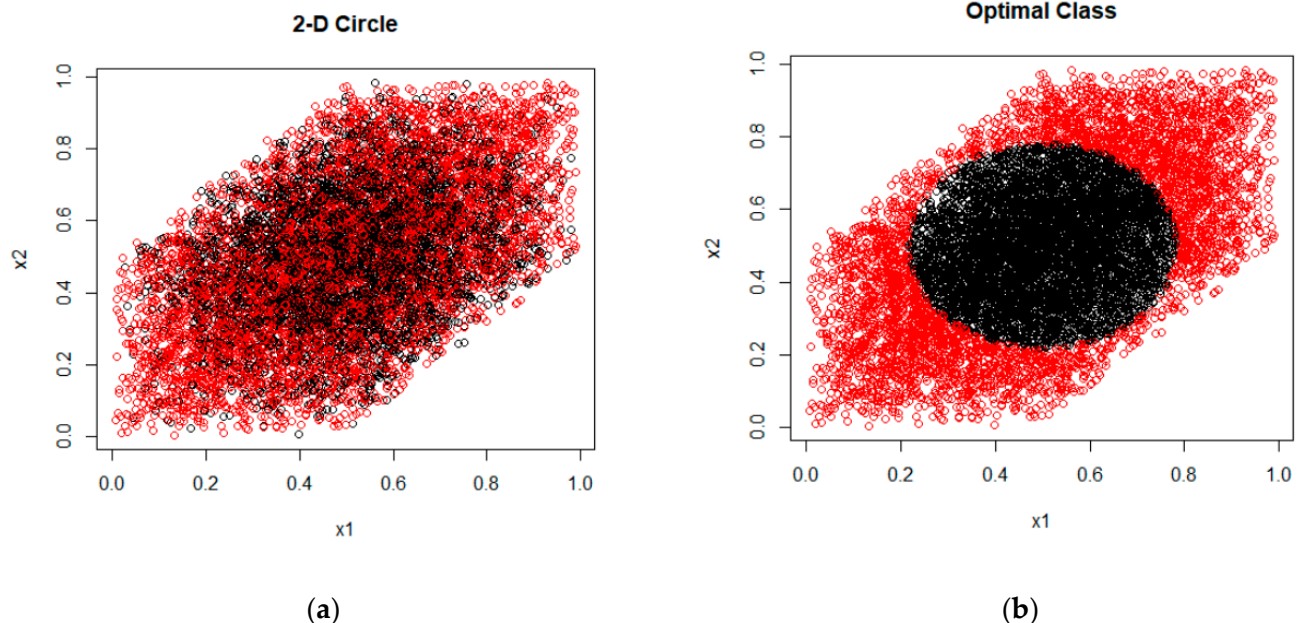

(**a**)　　　　　　　　　　　　　　　　　(**b**)

**Figure 3.** Class allocation for the 2-circle synthetic data set as a function of $x_1$ and $x_2$ showing: (**a**) The stochastic class labels; and (**b**) The correct classes that are used to find the optimal AUC.

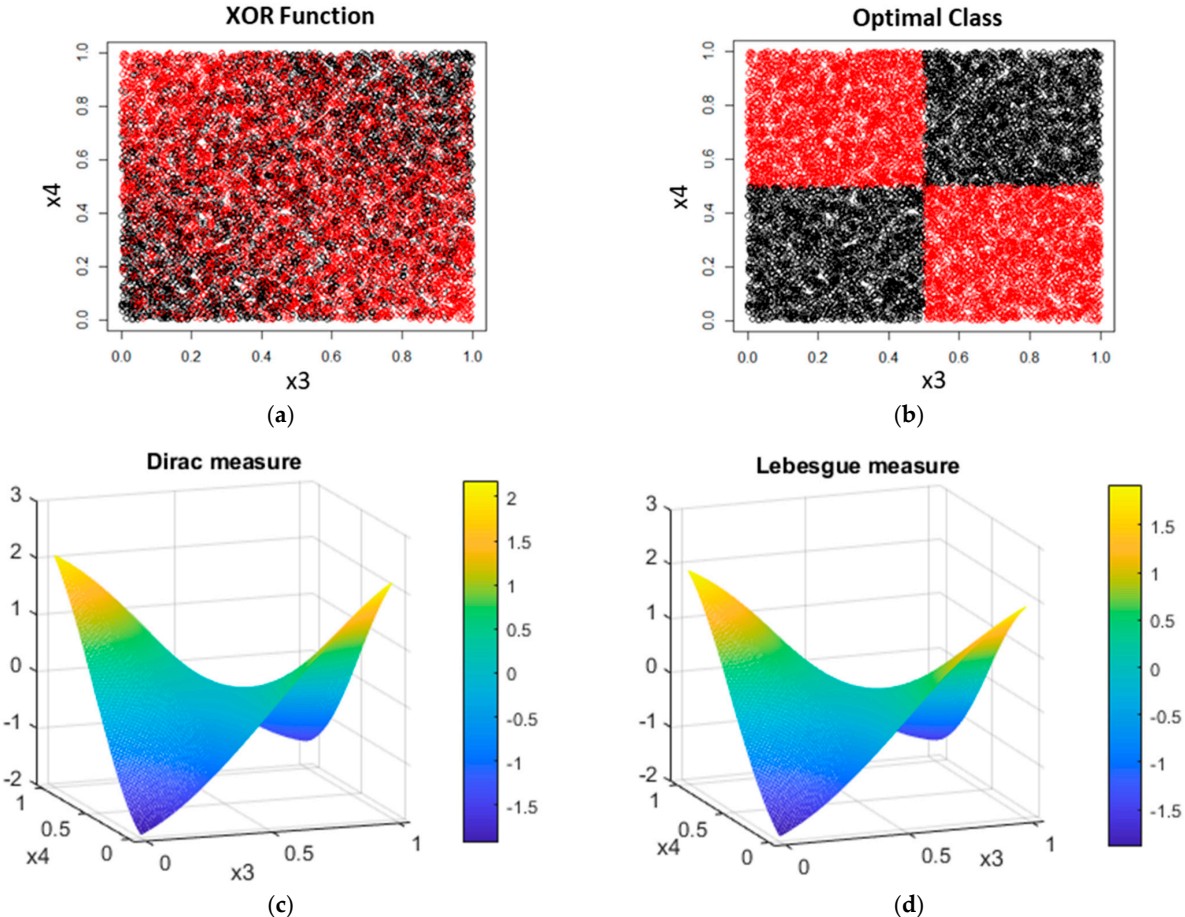

**Figure 4.** Class allocation for the XOR data set as a function of $x_3$ and $x_4$ showing: (**a**) The stochastic class labels; (**b**) The correct classes used to find the optimal AUC; (**c**) The two-way interaction term identified by the Dirac measure; and (**d**) The interaction estimated with the Lebesgue measure, which is almost identical to the curve in (**c**). Both surfaces are the only terms in the GAM, and closely correspond to the logit of the ideal XOR prediction surface. The main difference to theory is that the values at the four corners that saturate at finite values, whereas in theory, they extend to infinity in both vertical directions. This, however, has little impact on the crucial region for classification, which is the class boundary.

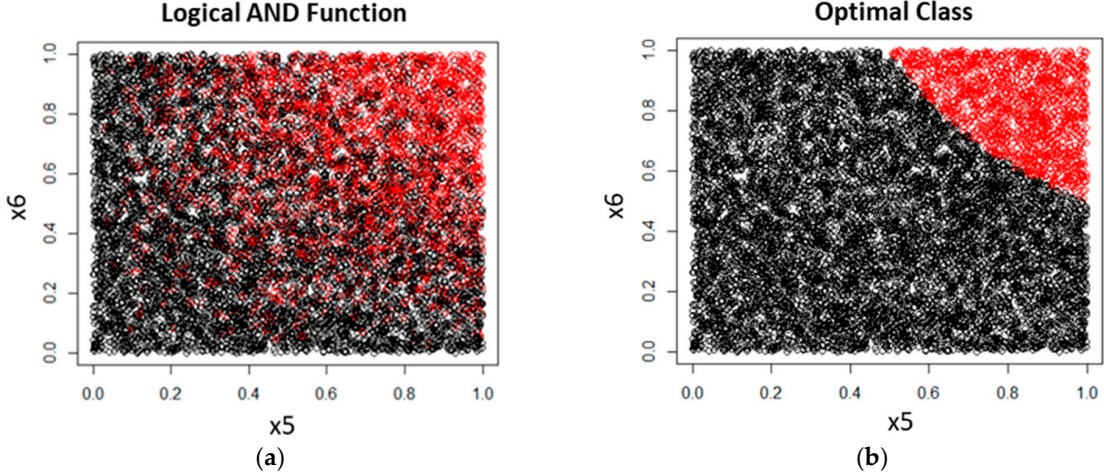

**Figure 5.** Class allocation for the synthetic data set representing the logical AND a function of $x_5$ and $x_6$ showing: (**a**) The stochastic class labels; and (**b**) The correct classes to find the optimal AUC.

The MLP-derived PRiSM models identified the correct ANOVA components for all data sets, with both the Dirac and Lebesgue measures. The three-way interaction term is a product of three inputs, not a pure interaction term. Therefore, its decision boundary can be approximated even in the absence of a third-order term, with three univariate partial responses sufficient to get close to the optimal prediction accuracy. Note that the predicted response in Figure 4c,d for the XOR task is very close to the bilinear surface corresponding to the theoretically correct response given by multilinear algebra (Tsukimoto, 2000).

Some machine learning models can be prolific in model selection with the ANOVA decomposition, followed by the Lasso, for either measure. The models always include the relevant variables but may also suffer from overfitting. However, it is remarkable how the interpretable models frequently achieve AUC values within 1% of the optimal value.

The models that filtered out the correct number of components to model each data set all have consistent interpretations. The partial responses correspond to the data generators in the vicinity of the class boundary, although the responses tend to level off away from the boundary, where the precise value of the logit is less important since the class membership probabilities are close to zero or one.

### 3.2. Real-World Data

The benchmarking results for the interpretable models against the original black box classifiers are summarised in Table 5. Values below the CI of the AUC are in italics.

**Table 5.** Classification performance for the real-valued data sets. The label 'D' indicates the number of input variables for the black boxes and component functions for the PriSM models.

| AUC [CI] | D | Diabetes | D | Credit Card | D | Shuttle |
|---|---|---|---|---|---|---|
| MLP | 7 | 0.902 [0.850,0.954] | 24 | 0.815 [0.758,0.872] | 6 | 0.999 [0.998,1.000] |
| SVM | 7 | 0.817 [0.749,0.884] | 24 | 0.793 [0.733,0.852] | 6 | 0.999 [0.999,1.000] |
| *GBM* | 7 | 0.816 [0.748,0.884] | 24 | 0.784 [0.724,0.845] | 6 | 1.000 [0.999,1.000] |
| **PRiSM models** | | | | *Dirac measure* | | |
| MLP–Lasso | 5 | 0.902 [0.851,0.954] | 12 | 0.818 [0.762,0.875] | 3 | 0.999 [0.999,1.000] * |
| PRN | 5 | 0.903 [0.851,0.954] | 12 | 0.809 [0.752,0.867] | 3 | 0.999 [0.998,1.000] * |
| PRN–Lasso | 5 | 0.903 [0.851,0.955] | 12 | 0.815 [0.758,0.872] | 2 | 0.998 [0.997,0.999] * |
| prSVM | 5 | 0.884 [0.829,0.940] | 13 | 0.798 [0.739,0.857] | 3 | 0.998 [0.997,0.999] * |
| prGBM | 8 | 0.847 [0.784,0.910] | 10 | 0.763 [0.700,0.825] | 2 | 0.998 [0.997,0.999] |
| **PRiSM models** | | | | *Lebesgue measure* | | |
| MLP–Lasso | 4 | 0.889 [0.835,0.944] | 12 | 0.819 [0.763,0.876] | 3 | 0.999 [0.998,1.000] * |
| PRN | 4 | 0.903 [0.852,0.955] | 12 | 0.817 [0.760,0.874] | 3 | 0.999 [0.998,1.000] * |
| PRN–Lasso | 4 | 0.905 [0.853,0.956] | 11 | 0.819 [0.762,0.875] | 2 | 0.999 [0.998,1.000] * |
| prSVM | 6 | 0.896 [0.842,0.949] | 12 | 0.803 [0.745,0.861] | 3 | 0.998 [0.997,0.999] * |
| prGBM | 7 | 0.881 [0.824,0.937] | 9 | 0.791 [0.732,0.851] | 2 | 0.997 [0.995,0.998] |

* Indicates a two-stage model selection process, explained in the text.

All methods use the same data sets, and the AUCs are quoted for test data only. Measuring statistical significance with the McNemar test shows that the performance difference between any pair of models is not significant at the 5% level.

While the accuracy of all models is comparable, the PRiSM models use fewer variables and are intuitive to interpret. It is also apparent that the two different measures lead to very similar classification performances. The coefficients of the Lasso used for re-calibration are close to unity for all models.

The number of component functions in Table 5 shows the effect of variable selection by the Lasso. The Diabetes data set generates only univariate responses. However, the Credit Card and Shuttle data sets require two-way interactions, as well as univariate effects. Note that the Credit Card data set generates 300 partial responses to choose from.

The GAMs, seeded by the SVM and GBM, are calibrated by the LASSO, resulting in the prSVM and prGBM. The univariate and bivariate structure of these models can be used

to define a PRN model, which is a SENN with MLP components, initialised either with random weights or with univariate and bivariate modules trained to replicate each of the selected partial responses. This will replicate the PRN and, following orthogonalization, the PRN–Lasso.

The sparsity of the models and their potential for interpretation are illustrated by the partial responses of two models, the MLP–Lasso and the PRN, shown in Figures 6–11. These functions are derived from the training data and are always used for prediction on out-of-sample data. The corresponding component functions for the other PRiSM models have similar values, although, if derived from random forests, e.g., in the case of the prGBM, they are stepwise constant rather than smooth. This is shown in [20] for a different data set.

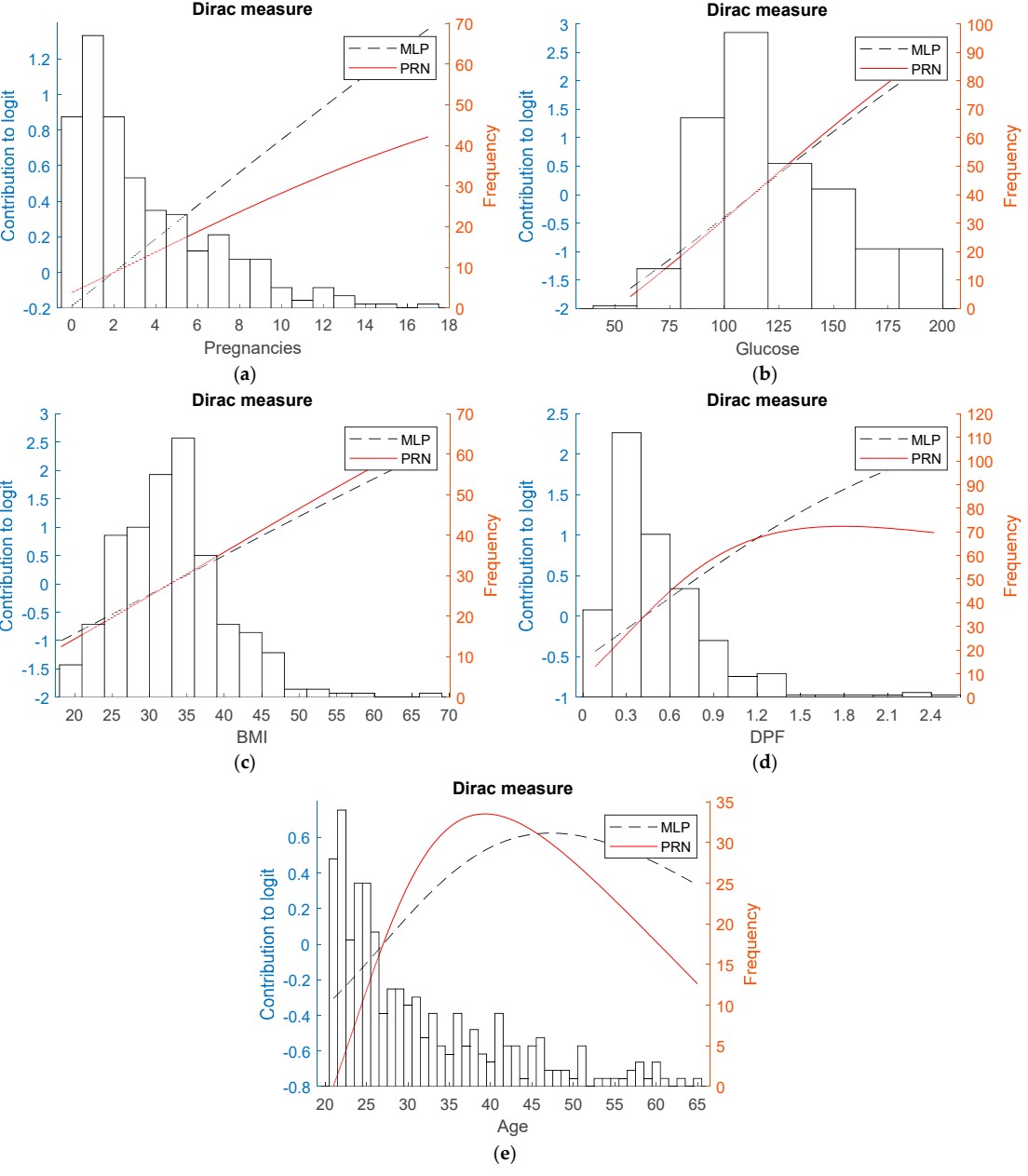

**Figure 6.** Contributions to the logit from partial responses to the logit (left axis) for the Diabetes data set, obtained with the Dirac measure, overlapped with the histogram of the training data (right axis). The final partial responses derived at the second application gradient descent (solid lines) are shown alongside the partial responses from the original MLP (dashed lines). Five covariates are represented, namely (**a**) Pregnancies, (**b**) Glucose, (**c**) BMI, (**d**) DPF and (**e**) Age.

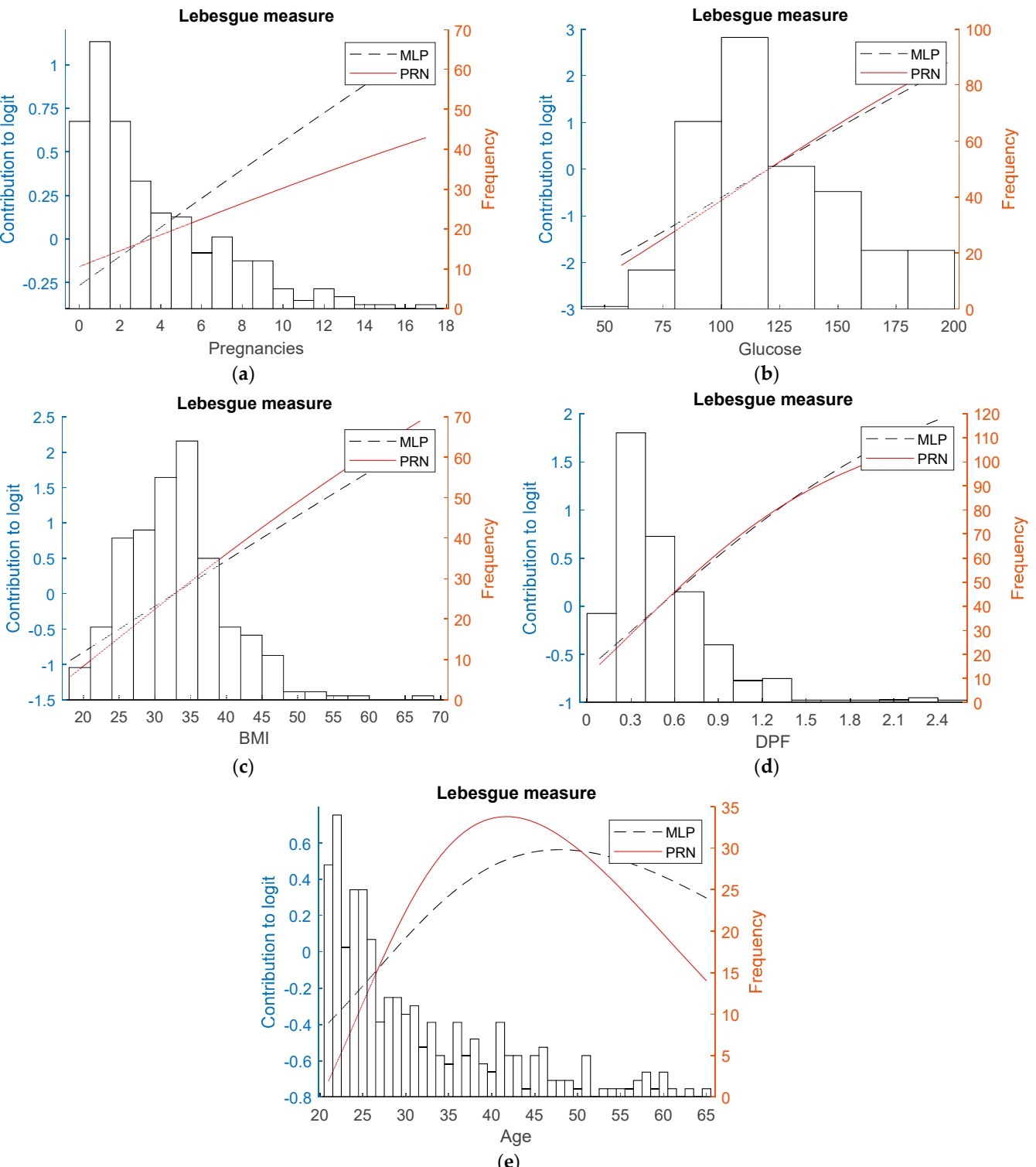

**Figure 7.** As for Figure 6, with the Lebesgue measure, the component functions of the GAM are very similar for both measures. They have a similar structure and range of contributions to the logit. Despite being fitted with a generic non-linear model, the MLP, several of the partial responses are linear. Variable "DPF" shows a saturation effect, as might be expected, while the log odds of "Age" as an independent effect peak around the age of 40. Note that data sparseness for higher values will result in greater uncertainty in the estimation of the partial response. The same size covariates are represented (**a**–**e**) as in Figure 6.

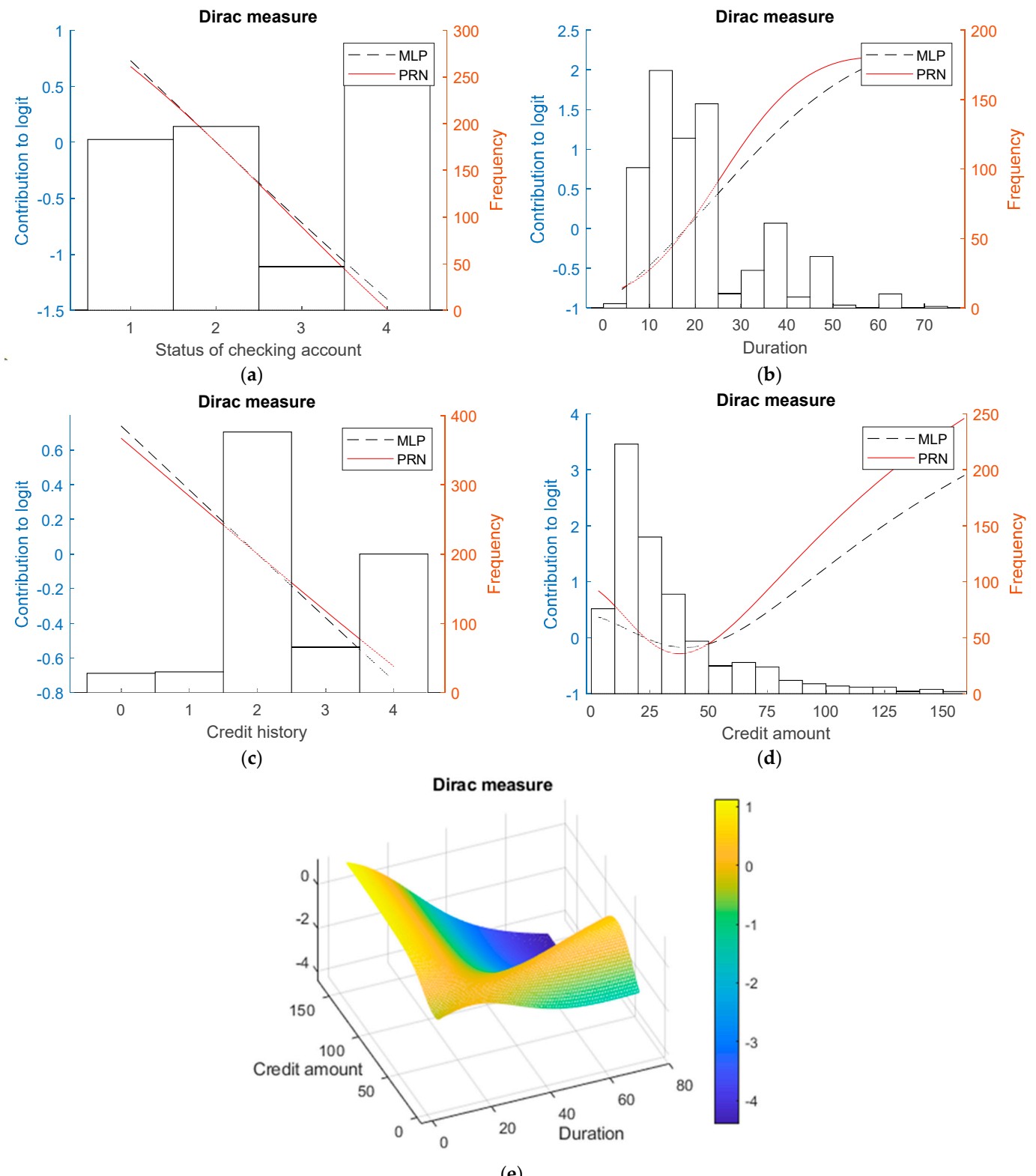

**Figure 8.** Partial responses for the German Credit Card data set, using the same notation as the previous figures. Four univariate responses and a bivariate response are shown namely for the covariates (**a**) Status of checking account, (**b**) Duration of loan, (**c**) Credit history and (**d**) Credit amount, together with (**e**) the pairwise interaction between Credit amount and Duration of loan.

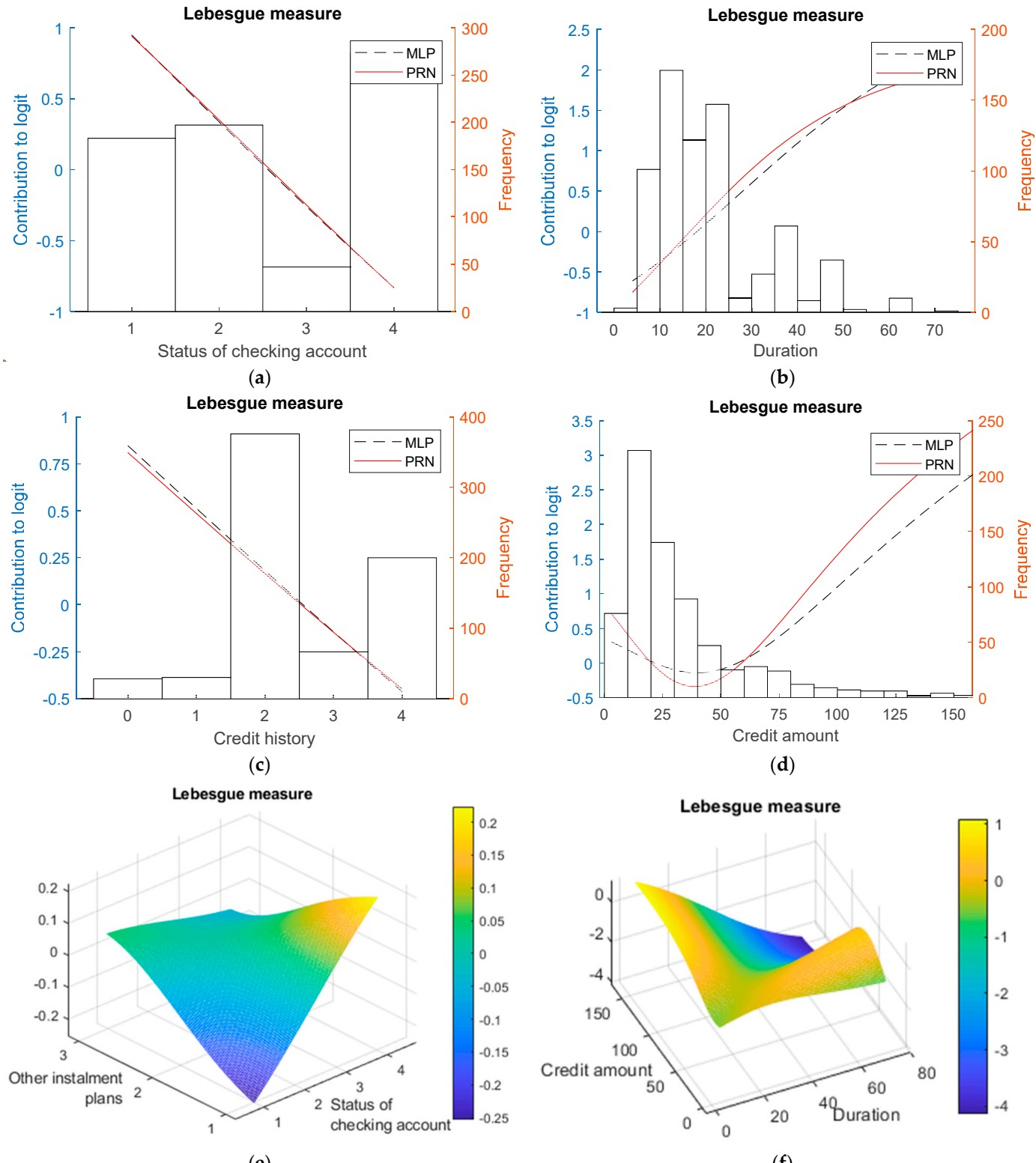

**Figure 9.** As for Figure 8, with the Lebesgue measure for the same covariates in (**a**–**d**), but with two pairwise interactions involving the variables listed in (**e**,**f**). Despite the different nature of the two measures, they offer entirely consistent interpretations, with the only difference being the selection by the Lasso model of a second bivariate interaction term, albeit with a range in contribution to the logit that is five times smaller than for the interaction term involving "Credit amount" and "Duration".

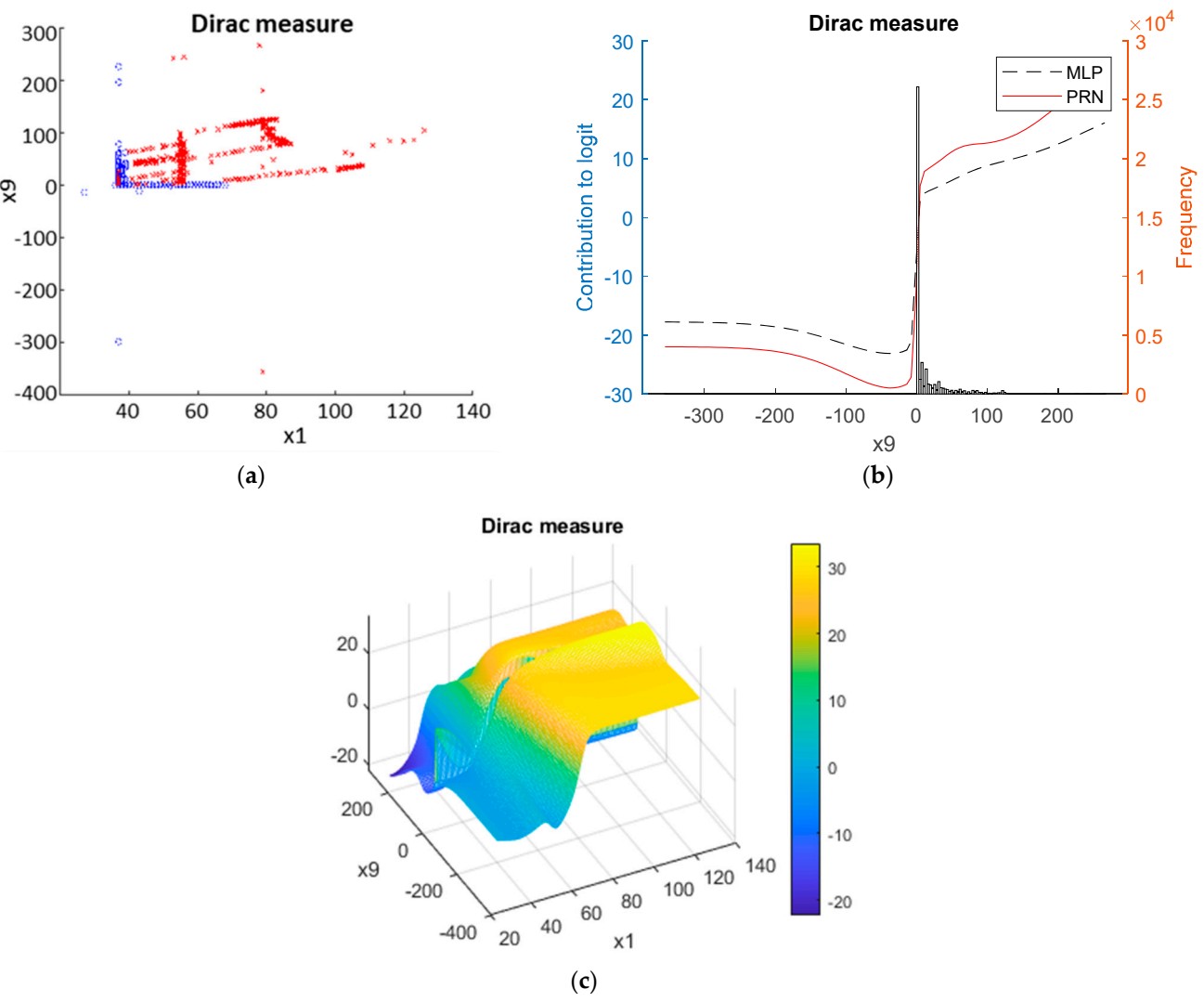

**Figure 10.** Nomogram of the PRN–Lasso model obtained for the Statlog Shuttle data set using the Dirac measure with a training/test split of n = 43,500 and 14,500, respectively: (**a**) Shows the raw data for the two variables selected, which corresponds well with two partial responses in the final model, namely: (**b**) shows the main effect involving $x_9$; and (**c**) plots the two-way interaction between the two variables in the model, $x_1$ and $x_9$.

Among the seven covariates in the Diabetes data set, five occurred together as univariate responses in all of the random initialisations for the MLP–Lasso, PRN–Lasso, and the two measures. They are "Pregnancies", "Glucose", "BMI", "DPF", and "Age". An interaction term involving "Glucose" and "DPF" was present in three random initialisations. The set of models obtained is, therefore, remarkably stable. The partial responses for the recurrent univariate effects are shown in Figures 6 and 7.

The German Credit Card data set is more challenging. Out of 24 variables, six were present in all initialisations for both measures: "Duration", "Credit history", "Savings accounts", "Period of employment", and the two variables labelled $x_{16}$ and $x_{17}$. In the case of the Lebesgue measure, three more variables recurred in all 10 initialisations, namely "Status of checking account", "Other instalment plans", and "Worker status". In addition, the variable "Credit amount" featured as a univariate or a bivariate term in eight initialisations. These ten variables were selected to obtain the models for which a selection of component functions is shown in Figures 8 and 9.

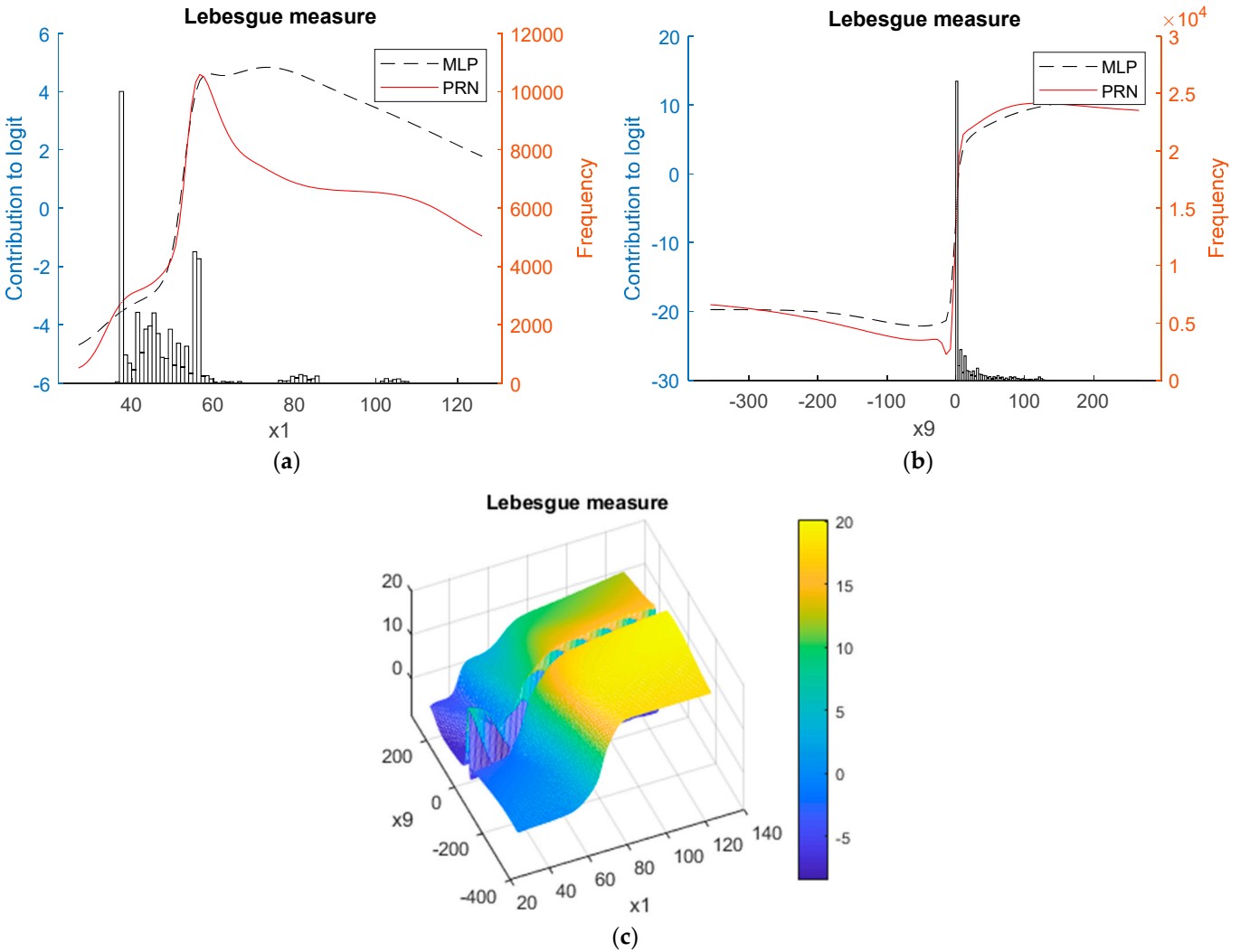

**Figure 11.** As for Figure 7, with the Lebesgue measure, the same two variables were used as with the Dirac measure, and similar AUC performance was achieved albeit involving an additional univariate term. Shown are the two main effects involving covariates $x_1$ in (**a**) and $x_9$ in (**b**) together with the pairwise interaction between them in (**c**).

Four univariate functions for multi-valued input variables are consistently monotonically decreasing and very close to linear, suggesting that these indicators are well calibrated as independent effects on credit risk, quantifying reductions in risk with rising input values. "Duration" shows saturation in its contribution to risk for large values, and "Credit amount" has a non-linear response with a minimum value. The bivariate responses suggest that to optimise the overall calibration of the model, adjustments are required in addition to the main effects. This includes a risk reduction when the "Credit amount" and "Duration" are both high, and a slight enhancement when either is small compared with the median value.

After mapping the structure derived with the Dirac measure from the prSVM onto the PRN–Lasso, good discrimination was achieved with an AUC of 0.812 [0.755, 0.869] with just ten univariate effects. They comprised the six variables identified also by the PRN, together with the variables 'Personal status', 'Property', and 'Other instalment plans'.

For both the Diabetes and Credit Card data, the other benchmarked algorithms, SVM and GBM, generally select more components than the MLP and have worse generalisation performance, as evident from Table 5. If the partial response models derived from each machine learning algorithm are mapped onto a SENN and further trained, then their

performance becomes similar for all of the models and they select consistent input variables, although some models will include additional ones.

The scalability and power of the method can be illustrated using the Statlog Shuttle data set. This data set is challenging because all of the variables have non-normal distributions, often with very peaked histograms, hence very small entropy. Two of the variables, $x_5$ and $x_9$, have a Pearson correlation of $-0.875$.

When applying MLP, the weight decay parameters estimated for variables $x_2$, $x_4$, and $x_6$ are noticeably larger than the others, indicating that these variables are less informative. They were, therefore, removed from the data. In the case of the Dirac measure, univariate component functions for $x_1$ and $x_9$ were selected by the PRN–Lasso with an AUC of 0.996 [0.994,0.998]. Selecting just these two variables as the inputs resulted in the performance listed in Table 5, involving a univariate effect for $x_9$ together with the interaction between $x_1$ and $x_9$. The Lebesgue measure behaved similarly but for the same Lasso selection procedure, and included also a univariate effect for $x_1$ albeit without an appreciable performance improvement.

The prSVM model selection process also followed a two-stage process, ending with the same two variables, $x_1$ and $x_9$, for both measures, each time involved in two univariate effects and a bivariate term. Interestingly, the prGBM model converged straight away on the two-component solution involving a univariate effect for $x_9$ and an interaction between $x_1$ and $x_9$ with the Dirac measure; with the Lebesgue measure, it converged on two univariate effects.

## 4. Discussion

A potential advantage of the PRN model over other PRiSM models in their current formulation is that the PRN allows for a second step of training, where the univariate and bivariate responses are re-estimated without the constraints of the ANOVA framework. The PRN–Lasso then applies the ANOVA framework to the refined univariate and bivariate models obtained by the PRN, followed by re-calibration, using logistic regression with $L_1$ regularisation.

We now turn to three key questions for explainable machine learning methods: the accuracy of the resulting models, their stability in model selection and their interpretability. In particular, accuracy and stability are critical requirements for any interpretable model.

### 4.1. Predictive Accuracy

The ability to model data depends primarily on the capacity of the machine learning algorithm to fit the data structure given the observational noise. Each of the methods shown is capable of fitting the benchmark data although some of the point estimates of the AUC are close to the confidence limit boundaries of the best performing methods. Therefore, the different methods have different efficiencies for modelling specific data sets. However, all of the models have comparable performance, which is uniformly high and with no evidence of an interpretability vs. performance trade-off.

### 4.2. Stability

The discussion of the empirical results shows that model selection is stable for multiple random initialisations of the MLP algorithm. This suggests that the PRiSM framework resolves two major limitations of the MLP: different predictions for multiple initialisations and lack of interpretation. It turns out that re-shaping the MLP to become a SENN also stabilises the predictions for different initialisations.

Stability is also good between models, with consistency between the input variables selected by all of the methods. This is perhaps most striking for the Shuttle data, where all of the methods picked the two key variables and identified an important interaction between them from a highly non-linear but remarkably noise-free set of measurements.

*4.3. Interpretability*

This can be referred to a formal framework involving the three Cs of interpretability [23]:

- Completeness—the proposed models have global coverage in the sense that they provide a direct and causal explanation of the model output from the input data, over the complete range of input data. The validity of the model output is evidenced by the AUC and calibration measures;
- Compactness—the explanations are as succinct, ensured by the application of logistic regression modelling with the Lasso. The component functions, both univariate and bivariate, are shown in the results to be stable, as are the derived GAMs;
- Correctness—the explanation generates trust in the sense that:
    - They are sufficiently correct to ensure good calibration for all data sets. This means that deviations from the theoretical curves for the synthetic data occur in regions where the model is close to saturated, i.e., making predictions close to 0 or 1;
    - The label coherence of the instances covered by the explanation is assured by the shape of the component functions so that the neighbouring instances have similar explanations.

The partial responses for the real-world data sets are plausible. In the case of the medical and credit card classifiers, some variables show remarkably linear dependence over their full range, while others are monotonic but their values saturate, showing a levelling-up beyond a certain point. In some cases, there is a turning point, and, interestingly, this was seen consistently for the two measures, e.g., for the added risk associated with credit amount, shown in Figures 8d and 9d. There is also a clear impact from data sparsity, which causes variability in the component functions in the less densely sampled regions of the data.

A more thorough appraisal of the plausibility of a PRiSM model using the Dirac measure applied to heart transplants is discussed in [21].

Finally, we note that PRiSM models are counterfactual because their predictions are directly connected to the input values. In the case of the PRN, the logit of the probabilistic prediction is simply the sum of the univariate and bivariate responses, whereas for the MLP–Lasso, PRN–Lasso, and the remaining PRiSM models seeded by other machine learning algorithms, the prediction is the sum of the response functions re-scaled by the linear coefficients of the Lasso.

## 5. Conclusions

We propose ANOVA decompositions of multivariate logit functions into sums of functions of fewer variables as a computationally efficient way to open probabilistic black box binary classifiers. Empirical results on the synthetic and real-world data show that the resulting interpretable models do not suffer from the interpretability vs. performance trade-off when applied to tabular data. Moreover, two alternative measures, Dirac and Lebesgue, lead to consistent interpretations for any given data set. The proposed method is accurate, stable, and scalable. Benchmarking it against a range of machine learning algorithms confirms this.

This paper formalizes the complete framework for the derivation of GAMs from black box classifiers, links the formalism to a commonly used attribution measure, Shapley values, and demonstrates its compliance with a user-led interpretability framework [23]. The paper extends previous results from related work focusing on clinical interpretations of a particular realization of the framework with anchored decomposition [20,21]. An unexpected finding of this study is that although the two measures are distinct, in that the Dirac measure represents a cut through a response surface at a particular point in the data and so is dependent on the choice of anchor point, while the Lebesgue measure integrates the surface over the range of the data and so is closer to the evaluation of size effects, both measures lead to similar interpretable models. This result is encouraging and suggests

that the PRiSM framework may be a viable method to derive globally interpretable models from arbitrary binary classifiers.

The resulting partial responses form a nomogram, which is a broadly used method of communicating complex models to users without the need for a detailed mathematical formulation [33]. We show that the components in the nomogram of a GAM agree exactly with the Shapley values, which are increasingly used for the explanation of machine learning in some high-stakes applications [34].

The component functions derived with the two measures, while close, are not identical. Further work is required to explore with end-users the preference for either measure and by which model they are seeded. In addition, confidence intervals for the univariate and bivariate terms need to be quantified. There are also clear parallels with regression models and a possible extension of the binary classifier to survival modelling within the framework of partial logistic cost functions [35]. Note that the method applies to any classifier that predicts the probability of class membership since it does not use the internal structure of the classifier but only the overall response function.

**Author Contributions:** P.J.G.L. conceptualised the method and led the study. P.J.G.L., S.O.-M. and I.O. implemented the code and ran the experiments. B.W. contributed to the discussions, implementation, and experiments. All authors evaluated the results. P.J.G.L., S.O.-M. and I.O. drafted the early versions of the manuscript. All authors contributed to the writing, reviewing, and editing, and approved the final manuscript. All authors have read and agreed to the published version of the manuscript.

**Funding:** This was partially funded by Liverpool John Moores University via a PhD scholarship.

**Data Availability Statement:** The real-world data analysed in the current study are available in Kaggle, the UCI Machine Learning repository, and PhysioNet, as follows: Diabetes (Pima Indians diabetes database): https://www.kaggle.com/datasets/uciml/pima-indians-diabetes-database (access on 11 October 2021) German Credit Card (Statlog German credit card data set): https://archive.ics.uci.edu/ml/datasets/statlog+(german+credit+data) (access on 11 October 2021). Statlog Shuttle data set: https://archive.ics.uci.edu/ml/datasets/Statlog+(Shuttle) (access on 11 October 2021). Medical Information Mart for Intensive Care (MIMIC-III): https://physionet.org/content/mimiciii/1.4/ (access on 11 October 2021).

**Conflicts of Interest:** The authors declare no conflict of interest.

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
