# Peer review of "How to Open a Black Box Classifier for Tabular Data"

_algorithms, doi:10.3390/a16040181_

Round 1

Reviewer 1 Report

The Authors presented the results of their research in an understandable way. The algorithms and methods used have been clearly described. Nevertheless, minor improvements need to be made before publication, namely:

1. The current title is very general and capacious. I suggest changing the title to be more specific, corresponding to the content of the manuscript.

2. Line 132: The acronym AUC appears for the first time in this line. There is no definition of this indicator. Please complete this.

3. Figure 1: In this figure, the first use of the acronym Lasso appears, then on line 182. It is not expanded until line 242. This should be the first time it is used. Please correct this.

4. Figures 4, 6-11: Illegible values on axes. Numerical values obscured by axis descriptions. Please correct these Figures.

5. Figures 6a and 6b: Vertical sections are missing from the histograms. Please correct these Figures.

6. Line 705: The Authors use the phrase methodology. In its most common sense, methodology is the study of research methods. I think the wording here should be method, algorithm or similar, but not methodology.

Author Response

  1. The current title is very general and capacious. I suggest changing the title to be more specific, corresponding to the content of the manuscript.

Response: The title has been made more specific to the content of the paper.

  1. Line 132: The acronym AUC appears for the first time in this line. There is no definition of this indicator. Please complete this.

Response: The AUC acronym is now explained in full.

  1. Figure 1: In this figure, the first use of the acronym Lasso appears, then on line 182. It is not expanded until line 242. This should be the first time it is used. Please correct this.

Response: The acronym Lasso is now explained the first time it is used, which is in the caption to fig 1.

  1. 4. Figures 4, 6-11: Illegible values on axes. Numerical values obscured by axis descriptions. Please correct these Figures.

Response: We have updated the figures to enhance their quality and ensure that all numeric values can be appropriately read.

  1. Figures 6a and 6b: Vertical sections are missing from the histograms. Please correct these Figures.

Response: We have updated these figures as well.

  1. Line 705: The Authors use the phrase methodology. In its most common sense, methodology is the study of research methods. I think the wording here should be method, algorithm or similar, but not methodology.

Response: The word “methodology” (lines 716 and 746) was replaced by “method”.

Reviewer 2 Report

Generally speaking, the article is well written. I would suggest an acceptance.

Author Response

Generally speaking, the article is well written. I would suggest an acceptance.

Response:  Thank you for your positive comment.

Reviewer 3 Report

The authors presents the interpretability and performance for the Multilayer Perceptron, Support Vector Machines and Gradient Boosting Machines applied to synthetic data and several real-world data sets, namely Pima Diabetes, German Credit Card and Statlog Shuttle from the UCI repository.

The idea is so interesting but their can be improved in some equation expressions such as eq (13) for ambiguities. 

And to make it easier for readers to follow the meaning of the paper, I suggest that the authors highlight important values ​​in the tables (1-5).

Thanks.

Author Response

The authors presents the interpretability and performance for the Multilayer Perceptron, Support Vector Machines and Gradient Boosting Machines applied to synthetic data and several real-world data sets, namely Pima Diabetes, German Credit Card and Statlog Shuttle from the UCI repository.

The idea is so interesting but they can be improved in some equation expressions such as eq. (13) for ambiguities.

Response:  The sentence introducing eq. (13) has been clarified.

And to make it easier for readers to follow the meaning of the paper, I suggest that the authors highlight important values ​​in the tables (1-5).

Response:  Optimal AUC values are now listed in bold and values below the confidence interval are in italics. This is also noted in the sentence in the text referring to tables 1-5.

Reviewer 4 Report

The authors proposed a method to understand machine learning models. They build a Generalised Additive Models (GAMs) by using an L1 regularised logistic regression. GAMs are derived using two alternative measures, Dirac and Lebesgue.

The authors demonstrate and compare interpretability and performance for the Multilayer Perceptron, Support Vector Machines and Gradient Boosting Machines applied to synthetic data and several real-world data sets.

Some questions and suggestions:

1. How to apply this method to a GNN?

2. Image resolution needs to be improved.

3. Writing needs to be improved, too many results in the main text, some should go to the appendix.

Author Response

The authors proposed a method to understand machine learning models. They build a Generalised Additive Models (GAMs) by using an L1 regularised logistic regression. GAMs are derived using two alternative measures, Dirac and Lebesgue.

The authors demonstrate and compare interpretability and performance for the Multilayer Perceptron, Support Vector Machines and Gradient Boosting Machines applied to synthetic data and several real-world data sets.

Some questions and suggestions:

  1. How to apply this method to a GNN?

Response: Yes, the method applies to any classifier that predicts the probability of class membership, since it does not use the internal structure of the classifier but only the overall response function.

This comment was added at the end of the paper.

  1. Image resolution needs to be improved.

Response: We have updated the figures to enhance their quality throughout the manuscript.

  1. Writing needs to be improved, too many results in the main text, some should go to the appendix.

Response: We appreciate that there are many results. Reading the results is important in order to understand the factual basis for the claims made in the main text, so we respectfully suggest keeping the tables.

The figures presented are a selection of all of the figures generated by the PRiSM models. The reason for presenting only a sample is indeed to avoid having too many results.

Round 2

Reviewer 4 Report

No additional comments. Thanks.